# SCalDA: Semantics-Calibrated and Diffusion-Enhanced Data Augmentation

**Shibo Lv** [1]   **Jianmin Jiang** [1]

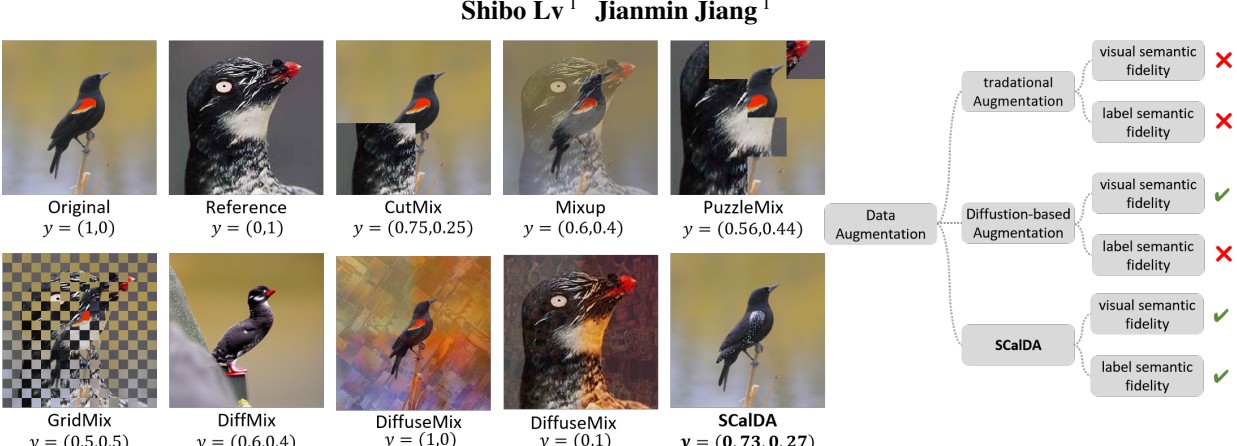

*Figure 1.* **Comparative illustration of the results achieved by our proposed and the existing baselines.** As seen, the existing methods (e.g., CutMix, Mixup) introduce unnatural artifacts and rely on linear label interpolation, failing to preserve both visual and label semantics. While diffusion-based baselines improve visual quality, they often suffer from label misalignment. In contrast, our proposed SCalDA achieves both **visual semantic fidelity** by generating realistic fused images and **label semantic fidelity** by assigning precise soft labels that correspond to the actual semantic content.

## Abstract

With the rapid development of deep learning, the issue of data scarcity has become increasingly prominent, inspiring emerging interests towards research on data augmentation techniques over recent years. However, our literature survey indicates that existing efforts often suffer from two issues of semantic infidelity, including: (i) visual semantics infidelity, such as visual artifacts, manifold intrusion, and unnatural blending boundaries etc, and (ii) label semantic infidelity, where augmented images do not match the original labels, creating extra label noises. To address these issues, we propose a Semantics-Calibrated and Diffusion-Enhanced Augmentation (SCalDA) scheme to achieve accurate semantics calibration across image, label and feature domains. Compared with the existing approaches, our proposed features in precise guidance in label domain, semantics driven synthesis across three domains (image, label and feature), and semantics-aware

metric learning. Extensive experiments on multiple datasets demonstrate that SCalDA yields consistent and significant performance improvements for both fine-grained and general classification tasks, validating the effectiveness and broad applicability of the proposed.

## 1. Introduction

Due to the rapid advances in computing hardware, the parameter scale is no longer a key bottleneck for deep learning. In modern network architectures, however, massive parameter counts still create problems for models to exceed their feature fitting capacity. Yet when the training data is limited, over-parameterized models could face a significant risk of overfitting (Zhang et al., 2017a). As a result, rich and high-quality data has become one of the key factors to forward the frontiers of learning performances (Sun et al., 2017). On one hand, constructing large-scale datasets such as ImageNet (Deng et al., 2009) can effectively alleviate this issue, on the other, various low-cost data augmentation techniques have been researched and published in the literature, opening up a new horizon for possible breakthroughs and solutions.

Among these data augmentation methods, Mixup (Zhang et al., 2017b) pioneered the use of linear interpolation and successfully smoothed the model's decision boundary.

[1]Shenzhen University, Shenzhen, China. Correspondence to: Jianmin Jiang <jianminjiang@szu.edu.cn>, Shibo Lv <2400101082@mails.szu.edu.cn>.

*Proceedings of the 43rd International Conference on Machine Learning*, Seoul, South Korea. PMLR 306, 2026. Copyright 2026 by the author(s).

Building on this, CutMix (Yun et al., 2019) introduced a region-level replacement strategy to further strengthen the model's perception of local features. Later, methods such as AugMix (Hendrycks et al., 2019) and PuzzleMix (Kim et al., 2020a) incorporated saliency priors, effectively overcoming the semantic loss incurred by random operations. Furthermore, with the surge of AIGC, He et al. (2022) validated the effectiveness of synthetic data in downstream tasks, and methods such as DiffMix (Wang et al., 2024), DiffuseMix (Islam et al., 2024), SaSPA (Michaeli & Fried, 2024) and SGD-Mix (Dong et al., 2025) significantly improved model performances by leveraging generated data in different ways.

Despite the remarkable progress made with the above methods, they still suffer from fundamental limitations in aligning visual semantics with label semantics. Fundamentally, the existing methods implicitly rely on a strong hypothesis that the pixel-area ratio equals the label-semantic ratio (Kim et al., 2020a), yet visual semantics are spatially non-uniform, and thus this hypothesis often does not hold in real-world scenarios. Moreover, these methods inevitably introduce hard boundaries or unnatural artifacts, damaging the natural image manifold. Although generative learning mitigates hard boundaries and manifold intrusion—a phenomenon where synthesized samples deviate from the natural data distribution and inadvertently overlap with the high-density regions of conflicting classes (Guo et al., 2019), their label assignment still depends on an imprecise assumption that the denoising process is linearly aligned with semantic changes. In other words, simply determining the label of synthetic data based on the number of steps at which new features are introduced is not accurate. As a result, further research is demanded to establish a reliable mapping between visual semantics and label semantics, which is a challenge for constructing high-quality synthetic data and developing corresponding augmentation techniques.

To address these challenges, we propose a SCalDA framework in this paper to sustain a unified semantic perspective via construction of three correlated representation domains, i.e. label domain, image domain and feature domain. To propagate label information into the image domain, we adopt saliency as the criterion for selecting labels and reconstruct a correlated region to characterize the exact position of a synthetic sample along the continuous semantic trajectory across these two domains. With such a bridging process, we employ a diffusion-based image inpainting model (Rombach et al., 2022) to perform local editing on those reconstructed parts, formulating a continuous and natural image domain to eliminate visual artifacts and manifold intrusion while aligning the semantics across both image and label domains. To further enhance the representational learning, finally, we introduce a semantic-aware metric-learning objective that explicitly transfers continuous semantic relations from label

domain to feature domain, constructing a geometric structure on the feature manifold that is consistent with the label and image domains. In this way, not only the label domain and the image domain are better leveraged with higher precision, but also the semantic alignment across all the three domains are better coordinated.

To validate the effectiveness of the proposed framework, we have conducted extensive evaluations on multiple datasets for image classification. The experimental results show that our proposed significantly outperforms existing SOTA baselines across these tasks.

**Conflict of Interest Disclosure.** The authors declare that they have no financial conflicts of interest related to this work.

## 2. Related Work

**Data augmentation.** Data augmentation aims to improve model generalization through geometric transformations (e.g., random cropping and flipping) (Shorten & Khoshgoftaar, 2019) or mixing strategies (e.g., Mixup (Zhang et al., 2017b) and CutMix (Yun et al., 2019)). However, random mixing strategies often ignore semantic content, leading to the loss of key regions and causing manifold intrusion (Guo et al., 2019; Harris et al., 2020; Gong et al., 2021). To resolve this issue, methods such as PuzzleMix (Kim et al., 2020a) and SaliencyMix (Uddin et al., 2020) introduce saliency priors to preserve highly discriminative regions, but they are essentially a mechanical splicing at pixel level, making it difficult to guarantee coherent visual context and precise alignment of label semantics (Huang et al., 2021; Chen et al., 2022; Kim et al., 2021; Dabouei et al., 2021). Guided by a precise label domain, we propose a new solution by constructing a joint mapping between the image domain and the feature domain, achieving more precise control over samples while ensuring both visual semantic fidelity and label semantic fidelity.

**Diffusion-based Augmentation.** With breakthroughs in diffusion models for high-fidelity image generation (Ho et al., 2020; Song et al., 2020; Dhariwal & Nichol, 2021; Rombach et al., 2022), generative data augmentation strategies have emerged to prevail the recent efforts towards high quality data augmentation (Trabucco et al., 2023; Azizi et al., 2023; Dunlap et al., 2023; Wang et al., 2024; Islam et al., 2024; Michaeli & Fried, 2024). He et al. (2022) reported their initial exploration in this direction and demonstrated its effectiveness, where a considerable domain gap between directly generated samples and real samples is reported. To overcome this shortfall, a range of attempts have been attempted by gradually shifting to image-guided editing strategies. DiffMix (Wang et al., 2024) performs an inter-class

image translation using a fine-tuned model to balance the diversity of generated samples and the fidelity of the foreground, whereas DiffuseMix (Islam et al., 2024) stitches the original image with generated variants to preserve key characteristics of the original input. Similarly, SaSPA (Michaeli & Fried, 2024) employs edge and subject conditioning to decouple generations from strict image guidance, whilst SGD-Mix (Dong et al., 2025) utilizes saliency-guided mixing to ensure foreground retention. While these methods mitigate visual distortions in traditional augmentation, they often overlook the fact that diffusion generations are not linearly correlated with semantic changes, inevitably introducing label noises. In contrast, our proposed performs precise region-level saliency computation, and applies image inpainting within a region to completely replace it with another category, thereby avoiding the above issues and achieving precise correspondence between the label domain and the image domain.

**Metric Learning.** Metric learning is the cornerstone of representational learning, aiming to build a discriminative feature domain by pulling samples from the same class closer and pushing samples from different classes further apart (Chen et al., 2020; Khosla et al., 2020). Therefore, metric learning has a natural advantage in distinguishing hard samples. Most existing supervised metric learning methods, however, only use discrete labels to define positive and negative pairs (Deng et al., 2019; Kim et al., 2020b). Although i-Mix (Lee et al., 2020) and MixCo (Kim et al., 2020c) take into account the continuous labels of mixup, the degree of their continuity is often approximated by the pixel mixing ratio, which is difficult to reflect real semantic changes. To this end, we propose to extract a precise region of the saliency from the generated images as the semantic coordinates, and then transfer the geometric structure to both label and feature domains to improve their ability of capturing features.

## 3. Method

As seen, SCalDA consists of three modules: (i) Saliency-based label alignment module, where a region-level saliency is extracted to determine an adaptive mixing ratio $\omega^*$. (ii) Diffusion-enhanced image alignment module, where an SDXL inpainting model is employed to blend the reference concept into the input image. (iii) Metric-based feature alignment module, which aligns the projected features of the synthetic image with the interpolated soft labels on a hypersphere, optimizing the contrastive objective.

### 3.1. Label Domain: Saliency-based Label Alignment

To address the label noise caused by inappropriate metrics in existing data augmentation, we propose to replace a specific region and leverage its "region saliency" as a guide for the synthesized label. To obtain the accurate saliency within this replaced region, we need to determine its precise region boundary. To this end, we adopt a region localizer (e.g., Grounded-SAM (Liu et al., 2024), (Kirillov et al., 2023)) to perform the region-level segmentation and build a coarse-to-fine semantic alignment framework, thereby constructing a high-precision label domain.

**Construction of Semantics Template:** To capture the structurally discriminative region in images, we construct a semantic template for each super-class within FGVC datasets. Considering the balance between the variety of different datasets and the sufficiency in covering the leading semantics inside every super-class, we define four semantic regions inside the template, which is described by texts and denoted as: $\mathcal{T} = \{t_k\}_{k=1}^K$. Taking CUB (Wah et al., 2011) as an example, whose super-class is birds, we define $K = 4$, leading to: $\mathcal{T} = \{\text{"head"}, \text{"breast"}, \text{"wing"}, \text{"tail"}\}$.

To establish the correspondence between the semantic region template and the input image, we inject the semantic region text $t_k$ into Grounding DINO for open-vocabulary detection. The corresponding bounding boxes produced by the detector are then used as spatial prompts for SAM, thereby generating a set of precise semantic region masks $\mathcal{M} = \{m_k\}_{k=1}^K$, where $m_k \in \{0, 1\}$ represents the pixel-level region boundary of the $k$-th anatomical region.

**Region Saliency Analysis:** To achieve high fidelity in data augmentation, we propose to calibrate the semantic importance in accordance with the level of their individual contributions across all $t_k$ inside the semantic template. Specifically, we use the heatmap $H$ generated by the saliency estimator (e.g., Grad-CAM (Selvaraju et al., 2017)) to mitigate the influence of background noises, where low-saliency filtering is adapted to obtain the purified heatmap $\hat{H}$. As a result, a semantic weight $\omega_k$ of the $k$-th region can be defined as:

$$\omega_k = \frac{\sum_{(u,v)} \hat{H}_{u,v} \odot m_{k,(u,v)}}{\sum_{(u,v)} \hat{H}_{u,v}} \tag{1}$$

where $m_{k,(u,v)}$ denotes the binary value of the $k$-th mask at position $(u, v)$. This ratio $\omega_k$ is designed to quantify the proportion of semantic information in the input image carried by the $k$-th region. The soft label for the synthesized image $y_{syn}$ can be determined as:

$$y_{syn} = (1 - \omega^*)y_{in} + \omega^* y_{ref}, \tag{2}$$

where $y_{in}$ and $y_{ref}$ denote the label vectors of the input image and the reference image, respectively. $\omega^* = \max_k(\omega_k)$ represents the maximum semantic weight among all candidate regions, which enables us to target the most salient region and extract complementary discriminative features

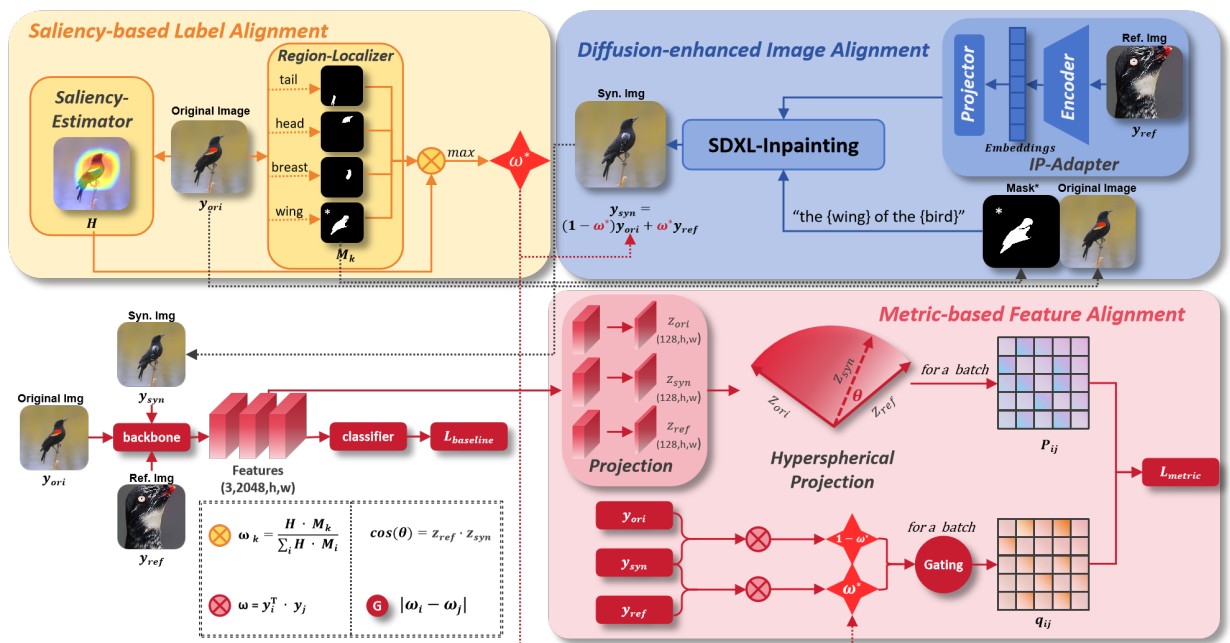

Figure 2. The overview of SCalDA framework.

from the remaining regions, thereby preventing overfitting to a single dominant feature. By targeting the most salient region, we effectively mitigate label noises and ensure precise transfer of semantic information, and thus achieve strict semantic alignment between the label and image domains.

### 3.2. Image Domain: Diffusion-enhanced Image Alignment

To overcome the infidelity problem incurred with the non-linear denoising in the existing diffusion-based approaches, we introduce an diffusion inpainting model (Podell et al., 2023) to generate a synthesized image across two different classes, where the alignment in label domain described in previous section is to be adopted as a guideline to complete the alignment inside the image domain. As a result, the synthesized image can be constructed by mask blending as described below.

For each input image $x_{in}$ and its saliency mask $m^*$, we first randomly select a reference image $x_{ref}$ from a different class, and construct a text prompt $\mathcal{P}$ combining the super-class of the input image with the targeted semantic region(e.g., The {wing} of the {bird}), and encode $x_{ref}$ into a high-dimensional visual feature embedding through IP-Adapter (Ye et al., 2023) $\phi_{ip}$. We then inject the text prompt together with the above image-related conditions into the diffusion inpainting model $\mathcal{G}$ to generate the synthesized image under the constraint of the mask $m^*$, details of which are given below:

$$x_{syn} = (1 - m^*) \odot x_{in} + m^* \odot \mathcal{G}(x_{in}, m^*, \phi_{ip}(x_{ref}), \mathcal{P}) \quad (3)$$

In this way, the overall structure of the input image is maximally preserved by the synthesized image, whilst a high fidelity region from the reference image is inpainted, ensuring that the accurate label ratio is not lost and thus achieving alignment between the image domain and the label domain.

### 3.3. Feature Domain: Metric-based Feature Alignment

To maximize the effectiveness of our proposed data augmentation, especially the diffusion-enhanced alignment across image and label domains, we need to sustain a feature alignment at the classifier end, in order to ensure that the best possible classification rate is achieved. To this end, we build a metric-learning branch on top of the semantic calibration framework as shown in Figure 2.

We first map the backbone features into a normalized low-dimensional space via a projection head, where $f(\cdot)$ adopts the standard MLP architecture (Chen et al., 2020) :

$$z = \frac{f(x)}{\|f(x)\|_2} \in \mathbb{R}^d, \quad \|z\|_2 = 1. \quad (4)$$

Given two images indexed by $i$ and $j$, respectively, we define the normalized feature similarity with respect to the other image as:

$$P_{ij} = \frac{\exp(z_i^\top z_j / \tau)}{\sum_{k \in A, \, k \neq i} \exp(z_i^\top z_k / \tau)}, \quad j \neq i, P_{ii} = 0. \quad (5)$$

where $A$ denotes the set of indices in the current mini-batch, and $k$ indexes all samples within $A$ except $i$, $\tau$ is a temperature parameter.

Let $y_i$ and $y_j$ denote the label vectors corresponding to image $i$ and image $j$. To achieve alignment between the feature and label domains, we propose to use the label similarity as the ground-truth. Formally, we define the similarity $S_{ij}$ between label $y_i$ and $y_j$ as:

$$S(i,j) = y_i^T y_j \tag{6}$$

As a matter of fact, this similarity between a synthetic label and its reference label is mathematically equivalent to $\omega^*$.

When aligning features within the set of synthesized images, however, we found that there often exist an overweighting problem when both synthesized labels are significantly different from each other. Under this circumstance, we further refine the semantic weight by introducing a constraining gate to mitigate the misleading supervision, and thus Equation 6 can be further reformulated into:

$$\tilde{S}(i,j) = (y_i^\top y_j) \cdot \exp\left(-\frac{\gamma}{\sqrt{2}}\|y_i - y_j\|_2\right) \tag{7}$$

where $\frac{\gamma}{\sqrt{2}}\|y_i - y_j\|_2$ is a scaled Euclidean distance between the two label vectors.

To align the label domain with the feature domain via normalized feature similarity described in Equation 5, we also normalize the label similarity via:

$$q_{ij} = \frac{\tilde{S}(i,j)}{\sum_{k \neq i} \tilde{S}(i,k)}. \tag{8}$$

where $q_{ij}$ stands for a normalized similarity between image $i$ and image $j$, and quantifies the strength of the supervision signal, thus transfers the semantic weight from the data augmentation to classification. As a result, the refinement effectively mitigates the risk of misleading supervision caused by matching images with incompatible information densities, ensuring that feature alignment is strictly driven by structurally consistent patterns.

Finally, to optimize the alignment between feature and label domains, we reformulate this alignment as minimizing the cross-entropy between the two normalized similarities, and thus a metric loss can be introduced as follows:

$$\mathcal{L}_{metric} = -\sum_{i \in A} \sum_{j \in A, \, j \neq i} q_{ij} \log P_{ij}. \tag{9}$$

Essentially, $\mathcal{L}_{metric}$ takes the precise soft-label similarity $q_{ij}$ as the target and drives the feature similarity to match

it, thereby explicitly aligning the geometric structure of the feature domain with the saliency-calibrated label domain. Consequently, the overall training loss can be formulated as:

$$\mathcal{L} = \mathcal{L}_{baseline} + \lambda \mathcal{L}_{metric}. \tag{10}$$

## 4. Experiments

To implement SCalDA, we utilize Grounding DINO (Liu et al., 2024) and Segment Anything Model (SAM) (Kirillov et al., 2023) with official pre-trained weights to extract precise region-level masks. To conduct saliency analysis, we incorporate Grad-CAM (Selvaraju et al., 2017) with a classifier(initialized on ImageNet-1K) pre-trained on the corresponding training split. Subsequently, Stable Diffusion XL (SDXL) (Podell et al., 2023) is employed for image generation.

### 4.1. Few-Shot Classification

For few-shot classification, *CUB-200-2011* (Wah et al., 2011) dataset is adopted, where few-shot subsets are constructed by randomly sampling $k \in \{2, 10, 20\}$ images per class, which were then utilized to generate $5\times$ the amount of synthetic data to form the training corpus. For hyper-parameters, we set $\lambda = 0.6$, $\tau = 0.05$, and $\gamma = 8$. We compare SCalDA with Vanilla, CutMix (Yun et al., 2019), Mixup (Zhang et al., 2017b), AugMix (Hendrycks et al., 2019) and PuzzleMix (Kim et al., 2020a) for traditional methods, DiffMix (Wang et al., 2024) for Diffusion based methods.

Table 2 lists all the experimental results, from which it can be seen that our proposed SCalDA consistently outperforms all baselines across different data regimes. Further analysis also reveals that, as the volume of real training data increases, the benefits yielded by the existing data augmentation diminish rapidly, since the impact of the diversity and the key of data augmentation, is gradually diluted. In contrast, SCalDA's advantage over DiffMix paradoxically widens from 2.85% to 3.20%. We attribute this to the "fidelity" of the synthetic data. As the number of real data grows, the model becomes increasingly demanding for the fidelity of synthetic images. Under this circumstance, the proposed SCalDA continues to provide precise supervisory signals even when data is relatively abundant, thereby maintaining consistent performance superiority.

### 4.2. Conventional Classification

To investigate the performance gain of SCalDA under full-shot settings, we first extend our evaluation to four fine-grained datasets: CUB (Wah et al., 2011), Aircraft (Maji et al., 2013), Cars (Krause et al., 2013), and Dogs (Khosla et al., 2011). Baseline methods encompass traditional meth-

*Table 1.* Top-1 accuracy (%) on fine-grained and general datasets, where '-' denotes methods tailored for fine-grained tasks.

| Dataset | Vanilla | CutMix | Mixup | AugMix | PuzzleMix | DiffuseMix | DiffMix | SaSPA | SGD-Mix | **Ours** |
|---|---|---|---|---|---|---|---|---|---|---|
| CUB-200-2011 | 86.64 | 87.23 | 86.68 | 86.98 | 87.06 | 86.19 | 87.16 | 86.92 | 87.66 | **88.17** |
| FGVC-Aircraft | 89.09 | 89.44 | 89.44 | 90.03 | 90.12 | 88.81 | 90.25 | 90.59 | 90.16 | **91.45** |
| Stanford Cars | 94.54 | 94.73 | 94.49 | 94.98 | 95.03 | 94.59 | 95.12 | 95.34 | 95.27 | **96.37** |
| Stanford Dogs | 87.48 | 87.59 | 87.42 | 87.28 | 87.76 | 87.39 | 87.74 | 87.69 | 88.01 | **88.42** |
| **Average(FGVC)** | 89.44 | 89.75 | 89.51 | 89.82 | 89.99 | 89.24 | 90.07 | 90.14 | 90.28 | **91.10** |
| CIFAR-100 | 76.33 | 76.80 | 76.84 | 75.31 | 80.38 | 82.50 | - | - | - | **83.18** |
| Tiny-ImageNet-200 | 57.23 | 56.67 | 56.59 | 55.97 | 63.48 | 65.77 | - | - | - | **65.89** |
| **Average(General)** | 66.78 | 66.74 | 66.72 | 65.64 | 71.93 | 74.13 | - | - | - | **74.54** |

*Table 2.* Comparative listing of Top-1 few-shot classification accuracy (%).

| Method | 2-shot | 10-shot | 20-shot |
|---|---|---|---|
| Vanilla | 28.18 | 67.16 | 78.46 |
| CutMix | 28.62 (+0.44) | 67.11 (-0.05) | 78.56 (+0.10) |
| Mixup | 31.98 (+3.80) | 68.32 (+1.16) | 79.01 (+0.55) |
| AugMix | 33.52 (+5.34) | 69.28 (+2.12) | 79.75 (+1.29) |
| PuzzleMix | 34.91 (+6.73) | 70.12 (+2.96) | 80.57 (+2.11) |
| DiffMix | 36.19 (+8.01) | 70.50 (+3.34) | 80.96 (+2.50) |
| **Ours** | **40.21** (**+12.03**) | **73.35** (**+6.19**) | **84.16** (**+5.70**) |

*Table 3.* Quantitative comparison of image quality on CUB. ↓ denotes lower is better, and ↑ denotes higher is better.

| Method | FID (↓) | IS (↑) | LPIPS (↓) |
|---|---|---|---|
| Mixup | 18.06 | 5.70 | 0.38 |
| CutMix | **6.15** | **5.91** | **0.28** |
| DiffMix | 10.94 | 5.59 | 0.34 |
| DiffuseMix | 42.89 | 5.99 | 0.32 |
| **SCalDA (Ours)** | **8.28** | **5.31** | **0.29** |

ods (CutMix, Mixup, AugMix, PuzzleMix) and recent diffusion-based methods (DiffuseMix (Islam et al., 2024), DiffMix, SaSPA (Michaeli & Fried, 2024), SGD-Mix (Dong et al., 2025)). All models compared utilize ResNet-50 as the backbone, which is pretrained with ImageNet-1K, and the input image size is 448 x 448.

The results are summarized in Table 1. As seen, on the Aircraft and Cars datasets, SCalDA achieves the best performance due to their highly rigid structural features, outperforming the second-best method by over 1%. On CUB and Dogs datasets, which exhibit higher texture complexity and background diversity, the improvement achieved by the proposed SCalDA is limited due to the increase of difficulty in image editing.

Beyond the fine-grained visual classification, we have also evaluated SCalDA on general datasets, including CIFAR-100 (Krizhevsky et al., 2009) and Tiny-ImageNet-200 (Deng et al., 2009). For these experiments, we employ the PreActResNet-18 (He et al., 2016) as the backbone network. The input resolutions are set to $32 \times 32$ for CIFAR-100 and $64 \times 64$ for Tiny-ImageNet, consistent with the standard protocols. To adapt SCalDA to these general datasets, we define four semantic regions for each super-class based on the 20 official ones of CIFAR-100, and apply a similar partitioning strategy to Tiny-ImageNet-200.

The results are summarized in Table 1. While the performance gains of SCalDA on general datasets are relatively modest, they remain consistent and effective. Compared

with fine-grained classification, general datasets are more demanding with inherent complexities and substantially larger data volumes, where images contained often have low resolutions and intricate backgrounds, making the reference features extracted via IP-Adapter less distinct. Nevertheless, owing to SCalDA's robust calibration capability on synthesized data, our method still achieves marginal yet steady improvements under these challenging conditions.

### 4.3. Analysis of Semantic Alignment across Domains

**Image Domain:Visual Fidelity.** To validate that the Image Alignment module effectively enhances the visual fidelity of those synthesized images, we evaluate the proposed against representative baselines in FID (Heusel et al., 2017), IS (Salimans et al., 2016), and LPIPS (Zhang et al., 2018) metrics under the full-data setting of the CUB dataset. We compute these metrics for synthesized images by Mixup, CutMix, DiffMix, DiffuseMix, and SCalDA. The comparative results are listed in Table 3.

Compared with DiffMix and DiffuseMix, as seen, SCalDA achieves significant improvements across FID and LPIPS. The lower IS observed in SCalDA does not stem from a deficiency in generation quality as IS inherently favors high-confidence for signal class. Rather, SCalDA deeply injects the semantics of reference image through region inpainting, resulting in images with richer cross-category features.

Although CutMix has exhibited excellent performances in preserving pixel-level realism, the unnatural artifacts introduced by hard cropping have limited its effectiveness, which

*Table 4.* For PuzzleMix, we choose the mixing ratio=0.5 for its best result.

| Method | Mixed Saliency (↑) |
|---|---|
| CutMix | 1.00 |
| PuzzleMix | 1.23 |
| **SCalDA (Ours)** | **1.41 ± 0.12** |

*Table 5.* Quantitative evaluation of SAE (↓). The results represent the mean and standard deviation across the test set.

| Method | SAE (% ↓) |
|---|---|
| CutMix (Baseline) | 69.74 ± 20.56 |
| PuzzleMix | 38.61 ± 10.26 |
| **SCalDA (Ours)** | **29.72 ± 6.88** |

will be demonstrated in the next section.

**Alignment between Label and Image Domains:** To verify the effectiveness of SCalDA in aligning the label and the image domains, we first adopt the Mixed Saliency (MS) metric (Kim et al., 2020a) proposed by PuzzleMix to evaluate the synthesized images and list the results in Table 4.

As seen, SCalDA achieves an improvement of $0.18$ over PuzzleMix. Since MS only captures the total amount of preserved saliency, it cannot reflect the quality of alignment between image and label domains. Consequently, we propose the following *Saliency Alignment Error(SAE)* to provide a more comprehensive measurement of performances, details of which are described below:

$$S_i = \sum H_i \odot (1 - m^*), \quad S_j = \sum H_j \odot m^*,$$
$$SAE = |\omega^* - \frac{S_j}{S_j + S_i}|, \quad (11)$$

where $S_i$ and $S_j$ denote the saliency amount within the corresponding region for input image $i$ and reference image $j$, respectively. $\omega^*$ refer to the semantic weight as defined in Section 3.1, and $m^*$ denotes the mask coresponding to $\omega^*$. For CutMix and PuzzleMix, $\omega^*$ corresponds to their specific label mixing ratios and $m^*$ denotes their generated binary masks.

The results are presented in Table 5, from which it can be seen that the proposed SCalDA performs best in comparison with CutMix and PuzzleMix.

**Alignment between Label and Feature Domains:** To quantify the alignment between the feature and label domains, we introduce the *Proportion Consistency Score* (PCS), for which the principle is that, for a synthesized image, its feature-domain proximity to the reference class should strictly correspond to its semantic weight $\omega^*$ defined in Section 3.1. Let $c_{in}$ and $c_{ref}$ denote the class centroids in the feature domain, we derive the feature predicted weight

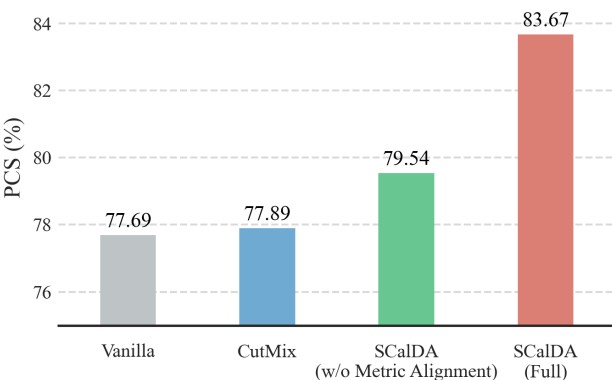

*Figure 3.* Evaluation of PCS.

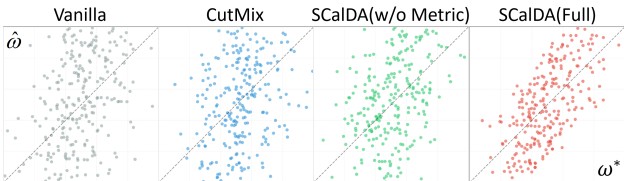

*Figure 4.* Scatter plots of the Label Semantic Proportion $\omega^*$ versus the Feature Proportion $\hat{\omega}$.

$\hat{\omega}$ via the softmax of cosine similarities:

$$\hat{\omega} = \frac{\exp(\cos(z, c_{ref})/\tau)}{\exp(\cos(z, c_{in})/\tau) + \exp(\cos(z, c_{ref})/\tau)}, \quad (12)$$
$$PCS = 1 - \mathbb{E}[|\hat{\omega} - \omega^*|],$$

where $\tau = 0.05$, $z$ denotes the feature of synthesized by ResNet-50 trained by different methods. A higher PCS indicates that the feature geometry linearly reflects the label interpolation. As shown in Figure 3, SCalDA achieves the highest PCS and exhibits a dense distribution along the diagonal $\hat{\omega} \approx \omega^*$. This confirms that the metric-based alignment module effectively enforces a linear transition in feature space consistent with semantic changes.

We further visualize the results using scatter plots in Figure 4, where 200 images are randomly selected for plotting. It is evident that SCalDA, equipped with the metric feature alignment, is significantly more concentrated along the diagonal line $\hat{\omega} = \omega^*$.

### 4.4. Ablation Studies

To quantify the contribution of each component within SCalDA, we conducted a progressive ablation study on the CUB dataset. The results are summarized in Table 6, based on which the following analysis is conducted.

**A Double-Edged Sword for Diffusion-enhanced Image Alignment (ID 1 vs. ID 2):** In ID 2, we replaced inpainting regions with random masks and area-based labels, leading

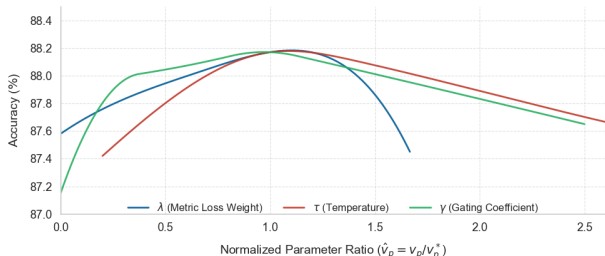

*Figure 5.* Parameter Sensitivity Analysis.To enable a comparative view across different scales, we plot the accuracy against the normalized parameter ratio $\hat{v}_p = v_p/v_p^*$, centered at the optimal setting ($v_p/v_p^* = 1$). The best values are $\lambda^* = 0.6$, $\tau^* = 0.05$, and $\gamma^* = 8$ respectively.

*Table 6.* Ablation study of SCalDA components on CUB, where each module contributes incrementally to the final performance, and $\Delta$ denotes the improvement over the Vanilla baseline (ID 1).

| ID | Baseline | Diffusion-enhanced Image Alignment | Saliency-based Label Alignment | Metric-based Feature Alignment | Acc(%) | $\Delta$ |
|----|----------|------------------------------------|-------------------------------|-------------------------------|--------|----------|
| 1 | ✓ | – | – | – | 86.64 | – |
| 2 | – | ✓† | – | – | 86.92 | +0.28 |
| 3 | – | ✓ | ✓ | – | 87.58 | +0.94 |
| 4 | – | ✓ | ✓ | ✓ | **88.17** | **+1.53** |

† denotes the use of random masking instead of saliency-guided masking.

to a marginal gain (86.64% to 86.92%), underperforming CutMix. This indicates that, without structural constraints, generative models may introduce semantic corruption and distribution shifts that outweigh diversity gains, which is in line with the findings of He et al. (2022).

**Effectiveness of Saliency-based Label Alignment (ID 2 vs. ID 3):** As seen in Table 6, integrating our saliency-based strategy (ID 3) boosts the accuracy to 87.58%, surpassing both the random baseline (+0.66%) and most diffusion-based baselines. This validates our hypothesis that precise semantic labeling outweighs texture diversity in generative augmentation. By preserving structural integrity and ensuring accurate region-level label ratios, we mitigate the negative impact of generation noises, thereby enhancing model discriminability.

**Feature Enhancement via Metric-based Feature Alignment (ID 3 vs. ID 4):** The inclusion of the Metric-based Feature Alignment module (ID 4) further pushes the accuracy to 88.17% (+0.59%). This result suggests that visual and label fidelity alone are insufficient, as synthetic features cannot measure the relation between its parents. By enforcing explicit feature-label constraints, SCalDA effectively utilizes saliency proportions to align synthetic features with their true geometric relationships, thereby maximizing the utility of synthesized images.

## 4.5. Parameter Sensitivity Analysis

To verify the robustness of SCalDA, we conduct a sensitivity analysis on three key hyperparameters: the metric learning weight $\lambda$ (Eq. 10), the temperature coefficient $\tau$ (Eq. 5), and the gating coefficient $\gamma$ (Eq. 7). All experiments settings are the same as Section 4.4, and the result is shown in Figure 5.

**Impact of $\lambda$.** The parameter $\lambda$ balances the primary classification loss and the Metric-based Feature Alignment. A small $\lambda$ ($< 0.2$) provides insufficient guidance for semantic alignment, while an excessively large $\lambda$ ($> 0.8$) hinders classification convergence. Optimal results are achieved within $\lambda \in [0.5, 0.8]$, effectively balancing feature alignment and classification accuracy.

**Impact of $\tau$.** The temperature coefficient $\tau$ regulates the concentration of the similarity distribution in the feature space. The result shows that a large $\tau$ ($> 0.1$) degrades accuracy due to over-smoothed distributions that obscure fine-grained features. Conversely, a too small $\tau$ ($< 0.03$) creates an excessively sharp distribution, disrupting feature manifold continuity. The optimal performance is achieved with $\tau \in [0.05, 0.1]$.

**Impact of $\gamma$.** When $\gamma = 0$, the gating mechanism is off, and the model fails to capture the smooth transitions of synthetic data, causing a clear performance drop. An excessive value of $\gamma$ also reduces accuracy as it weakens the connection between related images that have similar semantic weights. The range $[5, 10]$ yields the best performance.

## 5. Limitations and Conclusions

To address the critical issues of visual and label semantic unfaithfulness in existing augmentation methods, we propose a framework that aligns image and label domains, ensuring precise positioning within the interpolation manifold. We have also introduced a mechanism to transfer interpolation coordinates from labels to features, which effectively bridges the semantic gap. Extensive experiments demonstrate that this approach significantly enhances the model's fitting capability.

While SCalDA has delivered significant performance gains against a range of existing baselines, it is more computationally intensive due to those new modules added. However, this computational overhead can be mitigated by our design choices, which include: (i) We utilize efficient components such as Grad-CAM for saliency analysis and the optimized Grounding DINO-SAM combination for segmentation, which incur relatively low latency compared to the generation step. (ii) Unlike methods requiring per-class optimization (e.g., textual inversion or LoRA tuning), our adoption of the IP-Adapter allows for training-free feature injection, significantly reducing the deployment cost. (iii)

Synthesized images are generated offline, and thus it does not impede the training throughput of the classification model.

## Acknowledgements

The authors wish to acknowledge the financial support from Natural Science Foundation China (NSFC) under the Grant No. 62032015 and W2412099.

## Impact Statement

This paper presents work whose goal is to advance the field of Machine Learning. There are many potential societal consequences of our work, none of which we feel must be specifically highlighted here.

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

# A. Additional Experimental Details

*Table 7.* Detailed Semantic Region Templates $\mathcal{T}$ for FGVC datasets.

| Dataset | Semantic Region Keywords $\{t_k\}_{k=1}^{K}$ |
|---|---|
| CUB-200-2011 | {"head", "breast", "wing", "tail"} |
| Stanford Cars | {"headlight", "wheel", "door", "roof"} |
| FGVC-Aircraft | {"nose", "wing", "engine", "tail"} |
| Stanford Dogs | {"head", "leg", "tail", "body"} |

*Table 8.* Hyperparameters for Diffusion-Enhanced Image Alignment.

| Hyperparameter | Setting |
|---|---|
| Base Model | SDXL-Inpainting (v1.0) |
| IP-Adapter Weight | *0.8* |
| Inference Steps ($T$) | *25* |
| Guidance Scale | *7.5* |
| Upsampling Resolution | $512 \times 512$ |
| Target Resolution | $448 \times 448$ (FGVC) |
| | $32 \times 32$ (CIFAR-100) |
| | $64 \times 64$ (Tiny-ImageNet) |
| Saliency Filtering Threshold | $0.5 \times Avg(H)$ |

*Table 9.* Hyperparameters for classification.

| Hyperparameter | FGVC | General Datasets |
|---|---|---|
| Backbone | ResNet-50 | PreActResNet-18 |
| Optimizer | SGD | SGD |
| Momentum | 0.9 | 0.9 |
| Weight Decay | 5e-5 | 5e-4 |
| Initial LR | 0.02 | 0.1 |
| Batch Size | 64 | 128 |
| Epochs | 128 | 300 |
| Input Resolution | $448 \times 448$ | $32 \times 32$(C-100) |
| | | $64 \times 64$(T-IN) |
| *Metric Learning* | | |
| Metric Loss Weight $\lambda$ | 0.6 | 0.6 |
| Temperature $\tau$ | 0.05 | 0.05 |
| Gating Coefficient $\gamma$ | 8 | 8 |

# B. Semantic Templates for General Datasets

## CIFAR-100

**1. Aquatic Mammals:** {head, back, tail, belly}

**2. Fish:** {body, head, dorsal fin, tail fin}

**3. Flowers:** {petals, flower center, stem, leaves}

**4. Insects:** {wings, head, legs, body}

**5. Large Carnivores:** {head, body, legs, tail}

**6. Large Omnivores and Herbivores:** {head, body, legs, tail}

**7. Medium-sized Mammals:** {head, tail, legs, body}

**8. Non-insect Invertebrates:** {shell, legs, head, back}

**9. People:** {face, legs, body, arms}

**10. Reptiles:** {head, shell, tail, limbs}

**11. Small Mammals:** {legs, head, tail, body}

**12. Trees:** {crown, trunk, branches, leaves}

**13. Food Containers:** {rim, body, label, bottom}

**14. Fruit and Vegetables:** {skin, stem, flesh, side profile}

**15. Household Electrical Devices:** {screen, buttons, casing, cord}

**16. Household Furniture:** {surface, legs, backrest, cushion}

**17. Large Man-made Outdoor Things:** {roof, wall, windows, base}

**18. Large Natural Outdoor Scenes:** {sky, ground, vegetation, background}

**19. Vehicles 1:** {wheels, windows, body, headlights}

**20. Vehicles 2:** {cockpit, wheel, door, roof}

