# OpenReview forum: "SCalDA: Semantics-Calibrated and Diffusion-Enhanced Data Augmentation"
_ICML.cc/2026/Conference — ICML 2026 regular_

### Official Review · Reviewer_aKDz · 2026-03-01

**Soundness:** 3
**Presentation:** 3
**Significance:** 3
**Originality:** 3
**Overall Recommendation:** 4
**Confidence:** 4

**Summary:**

This paper introduces a data augmentation based on a diffusion-based image inpainting model. Unlike other methods, SCalDA considers the semantics calibration across image, label, and feature domains to obtain the semantic alignment data pairs. A wide range of experiments demonstrate that SCalDA significantly outperforms existing state-of-the-art baselines across these tasks.

**Compliance With Llm Reviewing Policy:**

Affirmed.

**Key Questions For Authors:**

- **(Q1)** Could the authors provide the computational overhead incurred by the generation stage? How does this overhead compare to other data augmentation methods in terms of time consumption and performance gains?

- **(Q2)** Could the authors provide experimental results on large-scale general classification datasets such as ImageNet-1K?

- **(Q3)** Could the authors provide an analysis or experiments about the semantic region keywords with the performance?

- **(Q4)** Could the authors add another backbone, like a ViT-based model, to emphasize the generalization?

- **(Q5)** Could the authors provide some visualizations?

**Limitations:**

- **(L1)** This method introduces three additional models, implying that its performance may depend on these models, thus limiting its generalization ability.

- **(L2)** The overhead incurred by the augmentation is unclear, especially in comparison with other augmentation methods.

**Strengths And Weaknesses:**

- **(W1)** This method relies on some extra models like a generative model for inpainting, a pretrain encoder for encoding images, and a SAM for segmentation. This brings an extra computational overhead. However, the authors did not provide a detailed analysis of time cost, training overhead, or GPU usage in the paper.

- **(W2)** The overall pipeline seems a little complex, which may limit the generalizability of the model. This is especially the case in scenarios with limited computational resources.

- **(W3)** Experiments on this method are currently limited to some fine-grained or relatively small-scale classification datasets (e.g., CIFAR100, Tiny-ImageNet). The paper does not provide results on standard large-scale, general-purpose benchmarks such as the ImageNet-1K dataset and benchmark.

- **(W4)** Experiments on the general datasets, it’s not clear whether the details of training, whether it’s online augmentation or offline by resizing the resolution to correct height and width.

- **(W5)** The authors reported that uesd some semantic region keywords for segmentation and masking, but the authors did not provided some responding analysis of them. And I recommend that the authors provide some visualization of the segmentation.

---

> ### Author Rebuttal · Authors · 2026-03-29
>
> Please find the details of our responses as follows
>
> **To Question-1:** **Points taken,** and the details of our added comparative analysis for the computational costs are given in Table-III and our response to Reviewer 6q3E.
>
> **To Question-2:** While the achievable improvement by SCaIDA could be limited for ultra large scale datasets, the core application for diffusion-based augmentation methods is essentially for scenarios with data-scare settings. As a matter of fact, the existing mainstream methods, such as DiffMix, SaSPA, and SGD-Mix, have never conducted any experiment on such datasets. Although DiffuseMix presented a strong performance on ImageNet-1K due to its far broader applicability than other existing models, its performance on the rest of the datasets, however, is very limited.
>
> As demonstrated by the experimental results in Table-2 of our original paper, under the extremely data-scarce 2-shot setting, SCalDA yields an improvement exceeding 12%, and as the amount of data increases, SCalDA continues to maintain stable performance gains. When the data volume reaches the scale of ImageNet-1K, however, the achievable gains by diffusion-based augmentation become extremely limited, and SCalDA has no exception. This would be an interesting direction for futher investigation in our future research.
>
> **To Question-3:** SCalDA does not depend on highly refined or strictly handcrafted templates, and as a matter of fact, its performance gains are primarily obtained from the semantic calibration framework itself. Even imprecise semantic templates can still provide semantic weight information beyond mere region area, supporting alignment in both the image and feature domains. This is validated by our experiments on general-purpose datasets, where SCalDA still achieves consistent performance improvements, despite the highly complex inter-class variation in CIFAR-100 and Tiny-ImageNet, which makes it difficult to provide sufficiently precise semantic templates. This fully demonstrate the robustness of SCaIDA to semantic template definitions.
>
> In addition, we are in the process of actively exploring the use of VLM models to automatically derive accurate semantic templates from datasets, addressing this issue with zero manual intervention. We will discuss this limitation in greater depth in the final version and provide a feasibility analysis, too.
>
> **To Question-4:** **Points taken.** We have supplemented few-shot and full-data experiments on the ViT-B/16 backbone. The results are listed in the following Tables:
>
> **Table A. Few-shot classification on CUB-200-2011 with ViT-B/16 backbone. Top-1 Acc (%)**
>
> | Method | 2-shot | 10-shot | 20-shot |
> | --- | --- | --- | --- |
> | Vanilla | 47.01 | 83.78 | 87.63 |
> | DiffMix | 54.25 (+7.24) | 85.44 (+1.66) | 88.56 (+0.93) |
> | SCalDA (Ours) | 56.97 (+9.96) | 86.94 (+3.16) | 88.69 (+1.06) |
>
> **Table B. Full-data FGVC with ViT-B/16 backbone. Top-1 Acc (%)**
>
> | Dataset | Vanilla | DiffMix | SGD-Mix | SCalDA (Ours) |
> | --- | --- | --- | --- | --- |
> | CUB-200-2011 | 89.37 | 90.05 | 90.44 | 90.78 |
> | FGVC-Aircraft | 83.50 | 84.33 | 85.21 | 85.97 |
> | Stanford Cars | 94.21 | 95.09 | 95.32 | 95.66 |
> | Stanford Dogs | 92.06 | 91.99 | 91.99 | 92.14 |
>
> The results demonstrate that SCalDA achieves consistent performance improvements on the ViT backbone as well, further demonstrating that SCalDA's effectiveness is indeed achieved by its semantic calibration framework rather than being tied to any specific backbone network. We have supplemented the experimental results in the revised version.
>
> **To Question-5:** **Points taken.** We have published some of the results in an anonymous repository for the reviewers' reference:https://anonymous.4open.science/r/Anonymous-5726/, and have included them in the appendix of the revised version.

---

> > ### Author Rebuttal · Reviewer_aKDz · 2026-04-01
> >
> > There is still the question of performance on large-scale datasets. It's more like a limitation on those diffusion-based augmentation methods. The authors said they will do some research in the future, so I decided to raise my score to 4.

---

> > > ### Author Response · Authors · 2026-04-07
> > >
> > > Appreciated.

---

### Official Review · Reviewer_aaGP · 2026-03-10

**Soundness:** 2
**Presentation:** 2
**Significance:** 3
**Originality:** 2
**Overall Recommendation:** 3
**Confidence:** 4

**Summary:**

This paper proposes SCalDA, a semantics-calibrated data augmentation framework that aligns label, image, and feature domains. The method first estimates a saliency-based soft label using region masks and Grad-CAM, then performs diffusion-based inpainting with SDXL and IP-Adapter, and finally imposes a metric-learning objective to align feature similarities with soft-label similarities.

**Compliance With Llm Reviewing Policy:**

Affirmed.

**Key Questions For Authors:**

(1) It is unclear whether the improvement comes from the semantic calibration mechanism itself or from the use of several strong pretrained modules. Without a clearer analysis of each module’s contribution and cost, the value of the proposed method is difficult to evaluate.

(2) Regarding the Region Saliency Analysis, the paper uses the maximum semantic weight among all candidate regions as the mixing coefficient for the input and reference images. This raises an important question: if only the most salient region determines the fusion weight, how are complementary discriminative features from the remaining regions actually captured and utilized?

(3)Regarding the diffusion model, is a pretrained model used in this work? If so, how is the conditioning incorporated? Please clarify how the conditional generation mechanism is implemented in the diffusion model.

(4) The paper should provide a clearer analysis of time complexity to better demonstrate the practical feasibility of the proposed method.

(5) The main results table currently reports only single-run Top-1 accuracy, while metrics such as SAE are presented with mean and variance. It would strengthen the credibility of the experiments to also report mean±std over multiple random seeds for the main classification results.

**Limitations:**

yes

**Strengths And Weaknesses:**

1 Strengths

The motivation is reasonable, the three-domain alignment story is coherent, and the ablations support the paper’s main claim that diffusion alone is insufficient without label and feature calibration.

2 Weaknesses

(1) I think that SCalDA primarily combines existing modules: Grounding DINO/SAM, Grad-CAM, SDXL inpainting, IP-Adapter, and a soft-label-driven metric loss. The overall system design is reasonable, but it is difficult to see a particularly core and concise new algorithmic idea in the paper. Therefore, from the perspective of ICML standards, its contribution appears closer to system integration than to a genuine methodological advance.

(2) The limitations section acknowledges that the method is more computationally intensive due to the introduction of multiple new modules, and argues that generation is performed offline and IP-Adapter is training-free. However, the main paper does not provide direct quantitative evidence, such as wall-clock time, GPU usage, preprocessing overhead, or storage cost. As a result, it is difficult for readers to assess whether the reported performance gains justify the added complexity.

---

> ### Author Rebuttal · Authors · 2026-03-29
>
> We address your questions regarding writing consistency, generalizability, semantic templates, and hyperparameter analysis point by point as follows:
>
> **To Question-1:** **Point taken.** We have corrected the caption of Figure 1 in the revised version and carefully proofread the entire manuscript to ensure consistency and accuracy throughout.
>
> **To Question-2:** **Point taken.** Due to the word limit of the abstract, we did not elaborate on how SCalDA addresses the two aforementioned problems. In the revised version, we have strengthened this part by added explanation, that we leverage the powerful generation capability of diffusion models to resolve the visual distortion (hard boundaries and visual artifacts) caused by naive stitching, and employ region-level saliency analysis to address the label distortion induced by area-based or denoising-step-based methods in other diffusion approaches.
>
> **To Question-3:** **Point taken,** and agree that validating a data augmentation strategy across multiple downstream tasks is highly valuable. As given in our response to Reviewer 6q3E's Q4, the core contribution of SCalDA lies in its semantic calibration framework itself: each component within the framework serves this central objective and is therefore inherently replaceable by design. To generalize SCalDA to other tasks, one can perform targeted substitutions of the corresponding components. For example, in object detection or semantic segmentation, the semantic weights in the label domain can similarly use the saliency weights of the most salient region, and the downstream classifier can be replaced with the corresponding detection or segmentation model.
>
> Representing a highly promising direction, we will adopt your valuable suggestion and actively pursue the relevant experiments, aiming to include preliminary results in the revised version.
>
> **To Question-4:** While relying on predefined semantic templates could be a limitation of SCalDA, we would like to address your concern in the following perspectives:
>
> - In real-world downstream tasks, encountering a dataset with absolutely no prior information that requires blind guessing is an extremely rare scenario. Researchers typically have a general understanding of the datasets they study, enabling them to define a usable set of semantic templates.
> - SCalDA is robust to the definition of semantic templates. As demonstrated by our experiments on highly diverse general-purpose datasets such as CIFAR-100 and Tiny-ImageNet, even under conditions of large inter-class variation and highly complex semantics, SCalDA still achieves consistent performance improvements with relatively coarse semantic templates defined for broad superclasses.
> - To address this limitation, we intend to incorporate a VLM module to summarize the common structures of objects and automatically generate suitable and accurate semantic templates without any manual intervention.
>
> We will discuss this limitation in greater detail in the final version and provide a conceptual analysis and feasibility discussion of VLM-assisted template generation.
>
> **To Question-5:** As stated in our paper, we adopted the same set of hyperparameters across all datasets without dataset-specific tuning, which validates that SCalDA's performance gains are achieved by the effectiveness of its semantic calibration framework, rather than careful hyperparameter selection. In addition, the sensitivity analysis on CUB further demonstrates that the model performance remains stable across a broad range of parameter values. For instance, reducing $\lambda$ to half its default value results in an accuracy fluctuation of only approximately 0.3%. This strongly indicates that SCalDA's performance gains are primarily attributable to its core semantic calibration mechanism, rather than a sensitive dependence on specific hyperparameter settings.
>
> **Table-IV: Main results with standard deviations (5 runs)**
>
> | Dataset | Top-1 Acc (%) Mean ± Std (5 runs) |
> | --- | --- |
> | CUB-200-2011 | 88.17 ± 0.27 |
> | FGVC-Aircraft | 91.45 ± 0.20 |
> | Stanford Cars | 96.37 ± 0.19 |
> | Stanford Dogs | 88.42 ± 0.25 |
> | CIFAR-100 | 83.18 ± 0.23 |
> | Tiny-ImageNet-200 | 65.89 ± 0.16 |

---

> > ### Author Rebuttal · Reviewer_aaGP · 2026-04-02
> >
> > After reading it carefully, I do not find that it addresses the questions raised in my review. I acknowledge that the authors added multi-run statistics, which is helpful. However, this only partially addresses my concern about result stability and does not resolve the main methodological and practical issues above.

---

> > > ### Author Response · Authors · 2026-04-07
> > >
> > > We would like to express our **sincerest apologies** for the mistake in our previous rebuttal, and we recognize that this was inappropriate and did not address your concerns. Below we provide a focused response to your questions.
> > >
> > > **To Question-1:Point taken.** We add a more fine-grained ablation study, and list the results in the following Table VII.
> > >
> > > **Table VII:Ablation Study.**
> > >
> > > | Setting | Description | Top-1 Accuracy (Mean ± Std) |
> > > | --- | --- | --- |
> > > | Base | Use CutMix to replace the predefined semantic parts, with the area ratio used as $\omega^*$. | 87.30 ± 0.14 |
> > > | Experiment-1 | Base + Saliency-based Alignment Module: replace the area ratio with saliency ratio as $\omega^*$. | 87.44 ± 0.12 |
> > > | Experiment-2 | Base + Inpainting-based Diffusion Module: replace the direct CutMix filling with diffusion-based inpainting. | 87.42 ± 0.16 |
> > > | Experiment-3 | Base + Metric-based Alignment Module: use the area ratio $\omega^*$ as the supervisory signal for the metric-based alignment module. | 87.28 ± 0.39 |
> > > | Experiment-4 | Experiment-1 + Inpainting-based Diffusion Module. | 87.58 ± 0.14 |
> > > | Experiment-5 | Experiment-1 + Metric-based Alignment Module. | 87.54 ± 0.35 |
> > > | Experiment-6 | Experiment-2 + Metric-based Alignment Module. | 87.92 ± 0.30 |
> > > | Experiment-7 | Full SCalDA. | 88.17 ± 0.27 |
> > >
> > > From these results, we can see that both the Saliency-based Alignment module and the Inpainting-based Diffusion module only make limited gains when added individually. In contrast, when the Metric-based Alignment module is introduced directly without preceding semantic calibration (Experiment-3), it not only fails to provide a stable improvement, but also exhibits noticeably larger variances. This is because $\omega^*$ defined by area ratio cannot accurately reflect the true semantic contribution, and therefore provides a noisy supervision for feature alignment. When the Metric-based Alignment module is combined with the other modules, however, improved effectiveness becomes noticeable.
> > >
> > > These results suggest that the final performance gain comes from the unified semantic calibration framework of SCalDA:Grad-CAM estimates the semantic weights in the label domain, SDXL-Inpainting aligns the image domain with the label domain, and Metric Learning aligns the feature domain with the label domain.None of these modules alone could possibly achieve such performance gains. We have included these improved experimental results in the revised version.
> > >
> > > **To Question-2:** Selecting the maximum semantic weight region only determines the replacement location and the mixing weight for a given image. This does not imply that the remaining complementary regions are neglected. On the contrary, the remaining regions continue to play a role at two levels: (i) all regions are jointly considered for the final determination of the maximum semantic weight region; (ii) as seen in Table-V of the second response to reviewer 6q3E, the unreplaced regions are preserved in the synthesized image, continuing to carry important discriminative information for the original class. As given in our response to Reviewer 6q3E's Q1, when the most discriminative region is replaced, the model must rely on the features of the remaining complementary regions to make correct judgments, thereby alleviating the excessive-reliance on a few local cues. Consequently, what the model ultimately learns is the holistic semantics jointly constituted by both the replaced and preserved regions, rather than the information from a single region.
> > >
> > > **To Question-3:** SCalDA uses a pretrained SDXL-inpainting model and does not require any additional fine-tuning of the diffusion model itself. Our method aims to perform local semantic region replacement while preserving the remaining semantics of the original image, and the SDXL-inpainting model is naturally well suited for this purpose. The model receives three types of inputs:
> > >
> > > 1. The regional mask provided by the Saliency-based Alignment module, which serves as the spatial condition and specifies the region to be replaced;
> > > 2. The text prompt (“the [region] of the [class] ”) provides the semantic description condition for the target region;
> > > 3. The reference image features provided by IP-Adapter supply the semantic and appearance conditions of the reference image.
> > >
> > > Since SCalDA only replaces a local region rather than generating a full image from scratch, these three inputs are sufficient for controllable and semantically guided inpainting without additional diffusion-model fine-tuning.We have clarified this implementation detail in the revised version.
> > >
> > > **To Question-4:Point taken,** and we have added comparative experiments to complete the complexity analysis, details of which are given in Table-III in our response to Reviewer 6q3E.
> > >
> > > We once again extend our sincerest apologies for this mistake, and sincerely appreciate your patience and thoughtful review.

---

### Official Review · Reviewer_3nUM · 2026-03-10

**Soundness:** 2
**Presentation:** 2
**Significance:** 3
**Originality:** 2
**Overall Recommendation:** 3
**Confidence:** 3

**Summary:**

The paper proposes a Semantic-Calibrated Diffusion Augmentation (SCalDA) method to address the problem of semantic inconsistency in existing data augmentation techniques, including visual artifacts and label noise. It achieves accurate semantic calibration across image, label, and feature domains, leading to improved performance in both fine-grained and general classification tasks.

**Compliance With Llm Reviewing Policy:**

Affirmed.

**Key Questions For Authors:**

See the Weakness.

**Limitations:**

The authors adequately discussed the limitations.

**Strengths And Weaknesses:**

Strengths:

1. By jointly modeling the **label domain, image domain, and feature domain**, the method achieves strict semantic alignment during the data augmentation process.

2. By leveraging **saliency-driven label calibration** and **region-level replacement strategies**, it ensures that the augmented images’ labels are consistent with their visual content, significantly reducing label noise and visual artifacts commonly observed in traditional augmentation methods.

Weaknesses:

1. The caption of Figure 1 is inconsistent, as it provides examples based on diffusion-based methods.

2. In the abstract, the logical connection explaining why the proposed method can address the two aforementioned problems is weak.

3. SCalDA demonstrates excellent performance in fine-grained classification, but it is unclear whether this three-domain alignment augmentation strategy is equally effective for downstream tasks such as object detection or segmentation.

4. The semantic region templates (e.g., “head, chest, wings, tail” for birds) are currently predefined for superclasses. For entirely new datasets lacking prior knowledge, it is unclear how the framework can automatically generate these semantic templates.

5. The metric learning weight $\lambda$ and temperature coefficient $\tau$ significantly impact performance. Although the paper provides a sensitivity analysis, it remains unclear whether these hyperparameters generalize across datasets of different scales, which is important to demonstrate the method’s effectiveness beyond engineering tuning.

---

> ### Author Rebuttal · Authors · 2026-03-29
>
> We address your questions regarding writing consistency, generalizability, semantic templates, and hyperparameter analysis point by point as follows:
>
> **To Weekness-1:** **Point taken.** We have corrected the caption of Figure 1 in the revised version and carefully proofread the entire manuscript to ensure consistency and accuracy throughout.
>
> **To Weekness-2:** **Point taken.** Due to the word limit of the abstract, we did not elaborate on how SCalDA addresses the two aforementioned problems. In the revised version, we have strengthened this part by added explanation, that we leverage the powerful generation capability of diffusion models to resolve the visual distortion (hard boundaries and visual artifacts) caused by naive stitching, and employ region-level saliency analysis to address the label distortion induced by area-based or denoising-step-based methods in other diffusion approaches.
>
> **To Weekness-3:** **Point taken,** and agree that validating a data augmentation strategy across multiple downstream tasks is highly valuable. As given in our response to Reviewer 6q3E's Q4, the core contribution of SCalDA lies in its semantic calibration framework itself: each component within the framework serves this central objective and is therefore inherently replaceable by design. To generalize SCalDA to other tasks, one can perform targeted substitutions of the corresponding components. For example, in object detection or semantic segmentation, the semantic weights in the label domain can similarly use the saliency weights of the most salient region, and the downstream classifier can be replaced with the corresponding detection or segmentation model.
>
> Representing a highly promising direction, we will adopt your valuable suggestion and actively pursue the relevant experiments, aiming to include preliminary results in the revised version.
>
> **To Weekness-4:** While relying on predefined semantic templates could be a limitation of SCalDA, we would like to address your concern in the following perspectives:
>
> - In real-world downstream tasks, encountering a dataset with absolutely no prior information that requires blind guessing is an extremely rare scenario. Researchers typically have a general understanding of the datasets they study, enabling them to define a usable set of semantic templates.
> - SCalDA is robust to the definition of semantic templates. As demonstrated by our experiments on highly diverse general-purpose datasets such as CIFAR-100 and Tiny-ImageNet, even under conditions of large inter-class variation and highly complex semantics, SCalDA still achieves consistent performance improvements with relatively coarse semantic templates defined for broad superclasses.
> - To address this limitation, we intend to incorporate a VLM module to summarize the common structures of objects and automatically generate suitable and accurate semantic templates without any manual intervention.
>
> We will discuss this limitation in greater detail in the final version and provide a conceptual analysis and feasibility discussion of VLM-assisted template generation.
>
> **To Weekness-5:** As stated in our paper, we adopted the same set of hyperparameters across all datasets without dataset-specific tuning, which validates that SCalDA's performance gains are achieved by the effectiveness of its semantic calibration framework, rather than careful hyperparameter selection. In addition, the sensitivity analysis on CUB further demonstrates that the model performance remains stable across a broad range of parameter values. For instance, reducing $\lambda$ to half its default value results in an accuracy fluctuation of only approximately 0.3%. This strongly indicates that SCalDA's performance gains are primarily attributable to its core semantic calibration mechanism, rather than a sensitive dependence on specific hyperparameter settings.

---

> > ### Author Rebuttal · Reviewer_3nUM · 2026-04-03
> >
> > I would like to thank the authors for their detailed responses, which have addressed several of my initial concerns. However, I still have a few remaining questions that require further clarification:
> >
> > 1. The current SCalDA pipeline relies heavily on Grounding DINO and SAM for precise part-level segmentation. If a VLM is introduced to automatically generate template vocabularies, how can we ensure that these generated text labels will be accurately recognized by pre-trained object detectors or segmentation models? Moreover, to what extent would the inherent inaccuracies in such recognition lead to cumulative errors throughout the subsequent semantic calibration process?
> >
> > 2. Object detection requires not only accurate classification but also precise bounding box (BBox) regression. The region-level replacement strategy in SCalDA inevitably causes significant shifts in both the image background and local object structures. Under such conditions, how does the framework automatically generate reliable and precise BBox annotations for the augmented images to support localized detection training?
> >
> > 3. Given the authors' claim that SCalDA outperforms DiffMix due to its superior "fidelity," how do you quantitatively evaluate the consistency between the generated "pseudo-local features" and the actual discriminative features of the ground-truth class? It would be helpful to provide evidence that the synthesized parts retain the fine-grained taxonomic characteristics necessary for accurate classification, rather than just appearing visually plausible.

---

> > > ### Author Response · Authors · 2026-04-07
> > >
> > > **To Question-1:** Our responses are highlighted as follows:
> > >
> > > 1. **Regarding cases where the templates cannot be recognized:** Even if a VLM is introduced in future work, we do not require all automatically generated templates to be recognized. Only when a template can be successfully localized and segmented by Grounding DINO and SAM in a specific image, it will be used in the subsequent semantic calibration process, and otherwise, it will not be adopted.
> > > 2. **Regarding cases where the recognition is not sufficiently precise:** SCalDA does not require highly precise semantic templates, and what it requires is just a reasonable semantic partition prior. This is supported by the stable gains on CIFAR-100 and Tiny-ImageNet-200, where the templates are relatively coarse.
> > > 3. **Regarding error accumulation:** We agree that inaccurate recognition may affect the subsequent semantic weight estimation and the effect of local replacement for that sample. However, this is better characterized as within-sample error propagation rather than cross-sample accumulation, due to the fact that each synthesized sample is processed independently.
> > > 4. **Regarding the role of VLM-based template generalization:** At the moment, using a VLM to generalize SCalDA for entirely new datasets is considered as part of our future research, since: (i) establishing new datasets is a process that requires enormous efforts and hence time-consuming; (ii) SCalDA has already been validated on most of the datasets adopted by the existing baselines, and better results have been achieved throughout the comparative experiments; and (iii) the superior performances delivered by SCalDA have demonstrated that the effectiveness of SCalDA is not dependent on those highly  precise semantic templates.
> > >
> > > Following your suggestion, we have added the above analysis into the revised version.
> > >
> > > **To Question-2: Point taken.** We agree that extending classification-oriented augmentation to object detection is non-trivial. As a matter of fact, this limitation is also shared by the existing methods,  such as PuzzleMix, DiffMix, DiffuseMix, SaSPA, and SGD-Mix. We therefore clarify this point in the revision and present it as part of our future work.
> > >
> > > **To Question-3:Point taken.** In response to this concern, we further supplemented an evaluation of DiffMix using the PCS metric, and the results are shown in the following Table VI.
> > >
> > > **Table VI. Comparison of PCS between DiffMix and SCalDA**
> > >
> > > | Method | PCS |
> > > | --- | --- |
> > > | DiffMix | 78.15 |
> > > | **SCalDA** | **83.67(+5.52)** |
> > >
> > > The results given in both Table VI and Table 3 in the original paper validate that  SCalDA outperforms DiffMix in terms of both visual fidelity and feature-label alignment.
> > >
> > > In addition, since DiffMix adopts global repainting and does not provide an explicit masked region, region-aware metrics such as MS or SAE cannot be directly applied for quantitative comparison.
> > >
> > > The comparison of DiffMix and SCalDA has been added into the revised version.

---

### Official Review · Reviewer_6q3E · 2026-03-12

**Soundness:** 3
**Presentation:** 3
**Significance:** 3
**Originality:** 3
**Overall Recommendation:** 5
**Confidence:** 4

**Summary:**

This paper proposes SCalDA, a framework for improving data augmentation by ensuring alignment across the image, label, and feature domains. Existing methods including both traditional and diffusion-based often suffer from visual infidelity and label inaccuracy. To address these issues, SCalDA develops a 3-module pipeline. It calculates semantic mixing ratio through region-level saliency, calculated using grad-cam and grounded-sam. Meanwhile, it uses SDXL inpainting model to smoothly mix concepts into the input image. For training, it adopts a metric-based objective to align the synthetic image features with the interpolated soft labels in the feature space. Experiments on multiple fine-grained classification datasets demonstrate improvements in few-shot and classic  classification tasks.

**Compliance With Llm Reviewing Policy:**

Affirmed.

**Final Justification:**

I am raising my score to an accept as the authors fully resolved my concerns during the rebuttal and follow-up discussions. The newly provided evidences fully validate the robustness and efficiency of the proposed framework.

**Key Questions For Authors:**

1. Would choosing maximal semantic weight component frequently yield a specific one of the four components?
2. What is the final label distribution of the augmented samples?
3. Does SCalDA involve a 2-stage training pipeline considering grad-cam? What is the time complexity?
4. Does SCalDA generalize to SDXL's OOD domains / highly specialized domains (e.g., medical imaging)?

**Limitations:**

yes

**Strengths And Weaknesses:**

Strength:

The paper is well-motivated to solve the alignment problems in synthetic augmentation images, with good integration of existing components that directly addresses the crucial flaw in current pipelines. The Saliency-based Label Alignment is sound in its method for utilizing heatmaps for label weight estimation. Meanwhile, the experimental results are promising and have clearly validated all the contributions.

Weaknesses:

- Choosing maximal semantic weight component ($\omega^*$) can frequently yield a specific one of them. E.g., wing for birds, due to its large area.
- The maximal weight strategy might also induce bias to the label distribution. It is unclear what is the final label distribution of the augmented samples and how they influence training.
- More case images are to be presented to illustrate the effectiveness of the proposed image domain approach, rather than only the one in Fig.1-2.
- Sec. 3.3 is unclear why there is an overweighting problem for two synthesized labels. Examples could be provided.
- The proposed method requires training, but relies on grad-cam which also relies on a trained model. This may make the pipeline time-consuming.
- The variable names are not aligned in figures and equations, e.g., Fig.2 and Eq.(2) $y_{in}$ vs. $y_{ori}$.

---

> ### Author Rebuttal · Authors · 2026-03-29
>
> Please find the details of our responses as follows:
>
> **To Question1 & Question2:** **Point taken.** We have collected statistics on which semantic parts are selected as the maximum semantic weight $(\omega^*)$ across datasets. The results are listed in Table-I as given below:
>
> **Table-I: Distribution of selected maximum semantic weight components across datasets**
>
> | Dataset | Part | Proportion |
> | --- | --- | --- |
> | CUB | wing | 36.13% |
> | CUB | breast | 31.89% |
> | CUB | head | 21.77% |
> | CUB | tail | 10.22% |
> | Aircraft | wing | 38.63% |
> | Aircraft | engines | 30.32% |
> | Aircraft | tail | 18.83% |
> | Aircraft | nose and cockpit | 12.22% |
> | Cars | side door | 40.93% |
> | Cars | front bumper | 27.86% |
> | Cars | rear bumper | 19.73% |
> | Cars | wheel | 11.48% |
> | Dogs | head | 42.37% |
> | Dogs | body | 27.15% |
> | Dogs | leg | 17.94% |
> | Dogs | tail | 12.54% |
>
> As seen, within each dataset, the maximum semantic weight indeed tends to fall on one or two local regions more frequently. This is consistent with the inherent nature of fine-grained recognition models, which typically rely on a small number of highly discriminative regions for decision-making. This is effective in most cases, but when inter-class differences are subtle, it may introduce recognition bias.
>
> To this end, we conducted a controlled experiment, i.e.: instead of selecting by the most salient region, we randomly select regions and distribute them as uniformly as possible across all parts. The results are shown in the following Table-II:
>
> **Table-II: Controlled experiment:**
>
> | Dataset | Top-1 Acc (%) |
> | --- | --- |
> | CUB-200-2011 | 87.96 (88.17) |
> | FGVC-Aircraft | 91.20 (91.45) |
> | Stanford Cars | 96.21 (96.37) |
> | Stanford Dogs | 88.26 (88.42) |
>
> As seen, across all four datasets, the accuracy under random uniform region selection is slightly lower than under the maximum semantic weight strategy. This indicates that uniform random selection fails to effectively mitigate the aforementioned over-reliance issue, resulting in a performance decline.
>
> The statistics and experimental results have been added in the revised version.
>
> **To Question-3:** **Point taken.** SCalDA does require a lightweight pretrained classifier to produce Grad-CAM maps. However, unlike DiffMix and SaSPA, SCalDA does not require any fine-tuning of the diffusion model itself, and thus the most expensive stage has been removed, compared with the existing diffusion-based augmentation pipelines. We have conducted a detailed comparison of the time costs of SCalDA, DiffMix, and SaSPA, details of which are listed in the following Table-III:
>
> **Table-III: Comparative listing of time costs**
>
> | Method | T_generate | T_train (Classifier) | T_total | Accuracy |
> | --- | --- | --- | --- | --- |
> | DiffMix | Fine-tuning: 2.4h / 4h (official repo) + Generation: 2h / 2.5h (official repo) = 4.4h | 0.7h | 5.1h | 87.16% |
> | SaSPA | 4.1h | 0.7h | 4.8h | 86.92% |
> | SCalDA | 0.2h + 2.7h = 2.9h | 1.0h | 3.9h | 88.17% |
>
> As seen, even accounting for classifier pretraining, SCalDA still reduces the total time cost from 5.1h (DiffMix) and 4.8h (SaSPA) to 3.9h, a reduction of approximately 20%, while achieving superior performances in terms of accuracy.
>
> These comparative costs have also been added into the revised version.
>
> **To Question-4:** **Point taken.** The core contribution of SCalDA is not a fixed combination of modules, but rather a semantic calibration framework, in which each module serves the goal of semantic calibration, and therefore they are inherently replaceable by design. For example, when applied to the medical imaging, SAM can be replaced with MedSAM (optimized for medical images), and SDXL can be replaced with domain-specific diffusion models such as RoentGen, thereby adapting to new domains while keeping the tri-domain alignment framework intact. We acknowledge that the current implementation, if directly applied to OOD or highly specialized domains, may not yield significant improvements. However, the replaceability of modules provides a clear path for cross-domain transfer, which is precisely the core value of SCalDA as a framework-level method. For your convenience of consideration, this point has also been made in the revised manuscript.

---

> > ### Author Rebuttal · Reviewer_6q3E · 2026-04-01
> >
> > The authors have mostly and excellently addressed the key concerns. However, it seems the rebuttal does not explicitly show the generated label distribution (not the part distribution): The maximal weight strategy might also induce bias to the label distribution. It is unclear what is the final label distribution of the augmented samples and how they influence training.
> >
> > I would raise my score if the authors can address this final concern.

---

> > > ### Author Response · Authors · 2026-04-07
> > >
> > > Following your suggestion, we further analyzed the distribution of $ω^∗$ over all synthesized samples. Since the final soft label is $\tilde{y} = (1-\omega^\ast) y_{ori} + \omega^\ast y_{ref}$, the distribution of $\omega^\ast$ directly reflects the final label distribution, as summarized in the following Table V.
> > >
> > > **Table V:Label distribution.**
> > >
> > > | Dataset | Total | 0–20% | 20–40% | 40–60% | 60–80% | 80–100% |
> > > | --- | --- | --- | --- | --- | --- | --- |
> > > | Aircraft | 6666 | 1120 (16.80%) | 2966 (44.49%) | 1400 (21.00%) | 827 (12.41%) | 353 (5.30%) |
> > > | CUB | 5994 | 1150 (19.19%) | 1870 (31.20%) | 1368 (22.82%) | 1030 (17.18%) | 576 (9.61%) |
> > > | Cars | 7726 | 1429 (18.50%) | 3183 (41.20%) | 1298 (16.80%) | 1287 (16.66%) | 529 (6.85%) |
> > > | Dogs | 8580 | 1888 (22.00%) | 3260 (38.00%) | 1698 (19.79%) | 969 (11.29%) | 765 (8.92%) |
> > >
> > > As seen, most $\omega^\ast$ values fall into the low-to-medium range, indicating that most synthesized samples are primarily dominated by the semantics of the original class, and the replaced region contributes only a small proportion of semantics rather than dominating the final mixed label. Meanwhile, a small portion of samples still exhibit relatively large $\omega^*$ values (e.g. on CUB), suggesting that local regions may in some cases carry strong class semantics.
> > >
> > > In the metric learning branch, we expect the feature space to form a smooth transition trajectory between the two classes according to the semantic proportion defined by $\omega^\ast$ in the label space. Intuitively, samples with different $\omega^\ast$  can be viewed as lying at different positions along this transition trajectory. If the supervision is mainly imposed on sample pairs with similar $\omega^*$, the model can progressively learn a smooth and well-ordered transition structure. In contrast, if equally strong constraints are also imposed on sample pairs with large semantic-proportion gaps, such pairs may introduce excessively strong supervisory signals, bending the trajectory, and thus damage its overall structure. Based on this consideration, we introduce the gating mechanism in Eq. (7) to attenuate the supervision from distant sample pairs while preserving the local attraction between nearby ones, thereby learning a smoother and more stable feature transition structure.
> > >
> > > This observation is also consistent with our hyperparameter analysis. As shown in Figure 5 of the main paper, when $\gamma = 0$, the gating mechanism becomes ineffective and the final accuracy drops by about 1%, which further provides experimental support for this design.
> > >
> > > **To Weakness-3:Points taken.** We have published some of the results in an anonymous repository for the reviewers' reference:https://anonymous.4open.science/r/Anonymous-5726/. Relevant details have also been included in the appendix of the revised version.

---

### Decision · Program_Chairs · 2026-04-30

**Decision:**

Accept (regular)

**Comment:**

The reviews are somewhat mixed, with one accept, one weak accept, and two weak reject recommendations. The paper addresses semantic inconsistency in data augmentation and proposes a unified framework that aligns label, image, and feature domains. Most reviewers find the motivation reasonable (Reviewer 6q3E, aaGP).

Reviewers 3nUM and aaGP raise concerns regarding generalization and novelty, including the reliance on predefined templates and whether the method primarily integrates existing components. In response, the authors clarify the modular and extensible nature of the framework and provide additional ablation studies and analysis. These responses partially address the concerns, although some questions remain open.

Overall, after carefully reviewing the paper and the rebuttal, the AC finds that while some limitations remain, they primarily relate to scope and do not undermine the core technical soundness of the work. Therefore, the AC recommends this paper for weak accept, as the strengths outweigh the weaknesses.